# Aberrant Splicing of *INS* Impairs Beta-Cell Differentiation and Proliferation by ER Stress in the Isogenic iPSC Model of Neonatal Diabetes

**DOI:** 10.3390/ijms23158824

**Published:** 2022-08-08

**Authors:** Alexandra V. Panova, Natalia V. Klementieva, Anna V. Sycheva, Elena V. Korobko, Anastasia O. Sosnovtseva, Tatiana S. Krasnova, Maria R. Karpova, Petr M. Rubtsov, Yulia V. Tikhonovich, Anatoly N. Tiulpakov, Sergey L. Kiselev

**Affiliations:** 1Vavilov Institute of General Genetics, Russian Academy of Sciences, 119991 Moscow, Russia; 2Endocrinology Research Centre, 115478 Moscow, Russia; 3Institute of Gene Biology, Russian Academy of Sciences, 119334 Moscow, Russia; 4Engelhardt Institute of Molecular Biology, Russian Academy of Sciences, 119991 Moscow, Russia

**Keywords:** diabetes, insulin, induced pluripotent stem cells, CRISPR/Cas9 genome editing, isogenic cell lines, proliferation, ER stress

## Abstract

One of the causes of diabetes in infants is the defect of the insulin gene (*INS*). Gene mutations can lead to proinsulin misfolding, an increased endoplasmic reticulum (ER) stress and possible beta-cell apoptosis. In humans, the mechanisms underlying beta-cell failure remain unclear. We generated induced pluripotent stem cells (iPSCs) from a patient diagnosed with neonatal diabetes mellitus carrying the *INS* mutation in the 2nd intron (c.188-31G>A) and engineered isogenic CRISPR/Cas9 mutation-corrected cell lines. Differentiation into beta-like cells demonstrated that mutation led to the emergence of an ectopic splice site within the *INS* and appearance of the abnormal RNA transcript. Isogenic iPSC lines differentiated into beta-like cells showed a clear difference in formation of organoids at pancreatic progenitor stage of differentiation. Moreover, MIN6 insulinoma cell line expressing mutated cDNA demonstrated significant decrease in proliferation capacity and activation of ER stress and unfolded protein response (UPR)-associated genes. These findings shed light on the mechanism underlying the pathogenesis of monogenic diabetes.

## 1. Introduction

Permanent neonatal diabetes mellitus (PNDM) is a type of diabetes that appears within the first months of life and persists throughout life. PNDM is caused by mutations in any of several genes, including the *KCNJ11*, *ABCC8*, and *INS* [1]. It can be inherited in an autosomal recessive or autosomal dominant fashion. PNDM is caused by the death or dysfunction of insulin-producing cells in the pancreas. Although insulin injections are used to replace this lost function, long-term complications occur. Transplantation of cadaveric allogeneic pancreatic islets containing cells has been successfully performed, demonstrating the feasibility of a cell therapy approach, which, however, is limited due to the small number of donors and the need for immunosuppressants [2]. Reprogramming an individual’s endogenous cells into induced pluripotent stem cells (iPSCs) with subsequent differentiation into disease-relevant cell types holds great promise for in vitro disease modeling, drug screening, and autologous cell replacement therapy for multiple diseases [3,4,5,6]. Pancreatic cells differentiated from iPSCs, which are obtained from diabetic patients, could be an excellent model for studying molecular mechanisms of diabetes-related diseases, as well as a source of autologous cell replacement therapy [7]. The most success in producing pancreatic beta-like cells from human PSCs has resulted from approaches that mimic normal pancreas development, which involves exposing the cells to various growth factors and signaling molecules at specific doses and in a particular sequence. To date, several approaches for PSCs differentiation into pancreatic beta-like cells have been developed [8,9,10,11,12,13]. Differentiated beta-like cells showed gene expression profile and other characteristics, including glucose-stimulated insulin secretion closely resembling beta cells found in pancreatic islets. However, in these multistage protocols, the final yield of the beta-like cells varied significantly, but hardly exceeded 50% leaving the majority of the remaining cells uncharacterized [14]. The genetic background of donors of iPSCs has a major impact on the differentiation behavior of the stem cells [15]. It complicates results interpretation since observed differences could be due to interindividual variability, but also to the specific genetic variant. Additionally, even successfully generated beta-like cells from PNDM patients will not be suitable for transplantation due to mutant genotype.

Development of engineered nucleases for site-specific DNA recognition and introduction of DNA breaks have greatly increased the feasibility of genome correction. Particularly, the CRISPR/Cas9 technology applied to human PSCs has gained popularity due to the unlimited life-span of the cells. The combined use of iPSCs and CRISPR/Cas9 DNA editing technology can reduce heterogeneity by generating isogenic cell lines that share the same genetic background and differ only in a genetic variant of interest. A number of studies that utilized this approach have been recently published to study neurodegenerative [16], metabolic [17], immunological [18], and endocrine [19] diseases. Recent studies utilizing CRISPR/Cas9 and iPSC approaches revealed some pathogenic mechanisms of neonatal diabetes [10,11], which are particularly connected with the insulin gene mutations previously studied in mice. ER stress in the in vitro differentiated beta-like cells was confirmed at gene expression and functional levels [12]. However, the mechanisms underlying diabetes manifestation for other heterozygous *INS* mutations still remain unknown.

To study the role of an intronic mutation, we established iPSC lines from a patient carrying the *INS* mutation in the 2nd intron (c.188-31G>A). The previous study suggested that this substitution creates an ectopic splice site, which results in a longer, abnormal transcript and presumably leads to the proinsulin misfolding followed by ER stress and beta-cell death [20]. CRISPR/Cas9 system was utilized to revert mutant cells to normal genotype. The existence of a pathogenic splice variant was demonstrated in the patient’s beta-like cells differentiated from iPSCs. Additionally, clear differences in formation of organoids at the pancreatic progenitor stage of differentiation were shown. However, genome editing and cell differentiation did not allow us to confirm the ER stress in the isogenic system. Using MIN6 insulinoma cell line expressing mutated cDNA, we demonstrated significant decrease in proliferation capacity and activation of unfolded protein response (UPR)-associated genes. Our findings suggest that PNDM-associated insulin mutation leads to the delayed development of beta-cell mass.

## 2. Results

### 2.1. Generation of iPSC-Based Isogenic System by CRISPR/Cas9 Gene Editing of the INS Gene

We generated an iPSC line from a patient with PNDM carrying an intronic mutation c.188-31G>A of the *INS* gene. Diabetes mellitus was diagnosed at the age of 7 months during a scheduled examination for glucosuria. The disease first manifested as hyperglycemia up to >14 mmol/L in the absence of beta-cell autoantibodies. No family anamnesis was found [21]. This MNDINSi001-A iPSC line was thoroughly characterized previously [22]. To generate isogenic control, we edited the disease mutation c.188-31G>A in the *INS* gene using CRISPR/Cas9 technology. Since the mutation is heterozygous, we targeted only the mutant allele by the allele-specific sgRNA. We selected three 20-nucleotide (nt) sgRNAs based on their on-target and off-target score calculation [23,24] (Figure 1A, Appendix A). Next, we evaluated the cutting efficiency of the designed sgRNAs in vitro in the patient’s iPSCs carrying the c.188-31G>A mutation. After transfection with CRISPR/Cas9-sgRNA plasmids (Geneart-sgRNA-1, Geneart-sgRNA-2, Geneart-sgRNA-3) the genomic DNA was isolated from sorted OFP-positive cells and analyzed by T7E1 assay. We did not find a significant difference between three sgRNAs, the cutting efficiency was 10%, 11.1%, and 8.9% for sgRNA-1, sgRNA-2, and sgRNA-3, respectively (Appendix A). For further studies, sgRNA-3 was chosen due to the close proximity of the mutation from the potential cut site [25]. The sgRNA-2 was used as a second choice. For homology-directed repair (HDR)-mediated incorporation of intended nucleotide ‘G’, the single-stranded repair oligodeoxynucleotide (ssODN) was designed, as well (Figure 1A). This 93-nt asymmetric ssODN contained a synonymous mutation in the first codon of exon 3 in the *INS* gene (GTG (Valine) → GTC (Valine) disrupting the BtsI restriction site to facilitate further clonal screening. 

MNDINSi001-A iPSCs were co-transfected with Geneart-sgRNA-2, 3 plasmids and ssODN by reverse lipofection. Despite the fact that the transfection efficiency is not very high (7–10%), this approach ensured more cell viability compared with electroporation (data not shown). To obtain single-cell clones, OFP-positive cells were sorted in the 96-well plates by FACS. To overcome the problem of high cell-death rate in single-cell culture, we tested different post-sorting conditions (Appendix A). The highest single cell viability was observed using feeder-based culture in mTeSR1, containing half of the conditioned medium supplemented with 10% KSR and Rho-associated kinase inhibitor. As a result, we obtained a total of 145 single colonies, which were picked and expanded for cryopreservation and further analysis.

First, we utilized restriction digest screening using BtsI enzyme, enabling us to rapidly analyze the HDR events in a cost-efficient manner. The target region containing c.188-31G>A mutation site was amplified from iPSCs genomic DNA and cleaved with BtsI followed by agarose gel electrophoresis. In the absence of ssODN sequence incorporation into Cas9 break site, we expected to see two bands (92 and 279 bp in size). An additional 371-bp band would indicate that HDR with ssODN as a template was successful. We analyzed 80 clones and 65 iPSC clones obtained with sgRNA-3 and sgRNA-2, respectively, and observed the expected ~400 bp fragment in only one clone (#19) (Appendix A).

Then, we performed Sanger sequencing of the target DNA region of 25 and 50 clones edited with sgRNA-3 and sgRNA-2, respectively (Appendix A). The cleavage efficiency for the sgRNA-3 and sgRNA-2 were 24% (6 out of 25 clones) and 22% (11 out of 50 clones), respectively (Figure 1B), which is consistent with data obtained for allele-specific CRISPR/Cas9 genome editing in human iPSCs [26]. Sequencing analysis did not confirm the HDR event in #19 clone, the additional band that was observed after restriction digestion was a result of an endogenous duplication. However, the expected correction (c.188-31 G>A) was found in two clones (#12 and #61) (Figure 1C). We conclude that the discrepancy between restriction digestion and sequencing data is most likely due to the fact that the homologous sister chromosome of wild type allele was used as a repair template rather than ssODN. The edited clone #61 (MNDINSi001-AE61) was chosen for differentiation into beta-like cells and further assays.

### 2.2. Differentiation of INS Mutant and Corrected iPSC Lines to Pancreatic Cells

Next, we differentiated unedited and edited cell lines (MNDINSi001-A and MNDINSi001-AE61, respectively), into pancreatic progenitor cells that express insulin to analyze the impact of the *INS* mutation on differentiation efficiency. H9 human ESC line was used as a gold standard for control. We failed to generate any NKX6.1/PDX1-positive cells by previously published protocol [8] and by commercially available Pancreatic Progenitor kit (Stem Cell Technologies, Vancouver, BC, Canada), though the lately published protocol [11] enabled us to generate NKX6.1/PDX1-positive cells from all three cell lines. For the 1st stage of differentiation, we used the Definitive endoderm kit. At day 5 after the induction of differentiation, cells were positive for Definitive endoderm (DE) markers: nuclear FOXA2, SOX17, and surface CXCR4 markers (data not shown). At day 14 of differentiation, cells in 2D culture were analyzed for pancreatic progenitor markers PDX1 and NKX6.1 (Figure 2A). We did not observe any significant differences in the percentage of the PDX1+ and NKX6.1+ cells between isogenic cell lines MNDINSi001-A and MNDINSi001-AE61 at this time point (Appendix A). Moreover, we analyzed the effectiveness of cell line differentiation into hormone-producing cells. At day 14 of differentiation, 2D culture cells were fixed and stained for the beta-cell hormones (insulin, C-peptide), alpha-cell hormone (glucagon), and delta-cell hormone (somatostatin) (Figure 2B). We did not observe any differences in the percentage of hormone-positive cells between unedited and edited cells at this pancreatic progenitor induction stage. Interestingly, there was a population of the multihormonal cells with simultaneous secretion of C-peptide and glucagon or C-peptide and somatostatin.

Differentiation into beta-cells proceeded for 7 more days using small molecules and growth factors to recapitulate pancreatic development by aggregating the final cell population into islet-like clusters. Surprisingly, cells differentiated from H9 and MNDINSi001-AE61 cell lines were able to form 80–200 μm organoids, wherein patient’s MNDINSi001-A cells formed the loose cell clusters with no compact round shape organoids (Figure 3A). We measured C-peptide level, a proinsulin-cleavage product that is co-secreted with insulin by pancreatic beta-cells, in culture media at days 14 and 21 using planar cell culture and organoids. In all culture conditions and time points, we detected C-peptide secretion (fresh cell culture medium served as a control), indicating that differentiated cells were capable of secreting beta-cell hormone during early differentiation events (Figure 3B). It is worth mentioning that patient’s beta-like cells repeatedly secreted lower levels of hormones, albeit it was difficult to evaluate the statistical difference due to the fact that the overall level of hormone secretion varied significantly between differentiation procedures.

### 2.3. INS c.188-31G>A Mutation Creates Insulin mRNA Isoform in Differentiated Human Beta-Like Cells without Insulin Production

We observed only minor differences between patient’s and edited cell lines during their differentiation at days 14 and 21 that could be possibly accounted for by the overall differences of two independently differentiated iPSC lines. Therefore, we decided to check the existence of the in silico predicted mRNA splice variant [20]. The NetGene2 server, a service producing neural network predictions of splice sites in humans, detected that the c.188-31G>A *INS* mutation leads to the new splice variant with the confidence 0.44 compared to 0.18 in the wild type allele (Figure 4A). Since differentiated mutant cells produced insulin, we decided to verify the existence of the predicted splice isoform along with wild type mRNA in the patient’s beta-like cells. We extracted RNA from isogenic beta-like cell lines and control H9 cells, converted to cDNA, amplified the region around mutation, cloned, and sequenced. In cDNA library from edited MNDINSi001-AE61 cells, we found only wild type *INS* transcript indicating that gene editing led to the homozygous restoration of the *INS* mRNA, whereas in cDNA library from patient’s beta-like cells there were clones containing the insertion of 29 bp between neighboring exons and the ones containing only wild type fragment of the mRNA (Figure 4B). Therefore, for the first time, we demonstrated that the c.188-31G>A *INS* gene mutation creates an ectopic splice site 30 bp upstream normal splice site of exon 3 of the *INS* gene.

Taking into consideration that wild type mRNA could be also expressed from the mutant allele along with splice variant, the resulting dose of the mutant transcript should be expected to be lower than the wild type. To compare the ratio of the wild type mRNA of the *INS* gene and the mutant one, we sequenced 32 *E. coli* clones from patient’s beta-like cells cDNA library. We found that 68% of clones contained wild type cDNAs, while 32% of clones contained mutant sequence with the 29 bp insertion. Therefore, for the first time, we demonstrated that even a negligible transcription level of the mutant insulin isoform results in detectable changes in insulin-producing cell development and hormone secretion in patient’s cells. To investigate whether splice variant isoform can be translated into insulin-like polypeptide in beta- and non-beta-cells, we transfected MIN6 and HEK293 cells with wild type and mutant cDNA constructs. Overexpression of wild type insulin under constitutive promoter resulted in a significant human insulin level detected by ELISA both in HEK293 and hormone secreting MIN6 cells (Figure 4C,D). However, we did not observe any insulin-like polypeptide synthesis from mutant isoform in both cell types. Some negligible signal was detected in MIN6 mouse insulinoma cells due to possible cross-reactivity with human anti-insulin antibodies.

### 2.4. Splice Variant Isoform Inhibits Proliferation of Mature Insulin Producing MIN6 Cells

Analysis of capability of patient’s and isogenic control iPSCs to differentiate into beta-like cells revealed only minor differences between them. This could be due to several factors including the complexity of the multi-stage process of differentiation, heterogeneity of differentiated cell populations, and rather low yield of beta-like cells. These observations might reflect pancreatic cell behavior in postnatal organisms when clinical manifestation occurs. Therefore, we decided to use insulinoma cell line MIN6 to study the gene isoform effects. We transfected MIN6 cells with fluorescently tagged wild type and mutant constructs. Efficiency of transfection was rather high allowing us to use fluorescent cells sorting of the transfected cells for further analysis. Collected cells were seeded on the Xcelligence analyzer plate and their proliferation was studied. First, we seeded cells on a 16-well Xcelligence analyzer plate 6 days after transfection, expecting that this time was enough for mutant transcript accumulation to affect the ER and cell growth. We detected slight but statistically significant loss of proliferation in cells expressing mutant transcript. Increasing the time up to 13 days for transcript accumulation allows us to observe substantial and statistically significant inhibition of growth of cells expressing mutant transcript (Figure 5A). Based on these data, it can be assumed that pathogenic c.188-31G>A mutation in the *INS* gene may lead to the loss of beta-cell mass in postnatal development.

### 2.5. ER Stress as a Possible Mechanism of Beta-Cell Dysfunction

Previously, it has been shown that coding mutations in the *INS* gene may affect insulin protein folding and secretion by ER stress and related unfolded protein response (UPR) presumably causing underdevelopment of the pancreas in PNDM [27]. Disruption of ER homeostasis leads to the accumulation of unfolded proteins. The ER has developed an adaptive mechanism called the UPR to counteract compromised protein folding. We decided to investigate the role of UPR as a possible mechanism involved in proliferation inhibition in the case of intronic c.188-31G>A mutation. We used ER Stress Antibody Sampler Kit, that contains a pool of molecular chaperone proteins including calnexin, GRP78/BiP, PDI, CHOP, etc. to analyze protein extracts from beta-like cells differentiated from isogenic iPSC lines and MIN6 cells transfected with wild type and mutant cDNAs. We did not find any differences between wild type and mutant alleles by Western blotting (data not shown). Therefore, we decided to use the more sensitive and quantitative luciferase reporter assay, where luciferase expression was controlled by GRP78/BiP promoter extensively used to indicate the onset of the UPR [28], unfolded protein response element (UPRE) or ER stress response element (ERSE) [29]. MIN6 cells were co-transfected with reporter plasmid and wild type or mutant constructs. Overexpression of wild type and splice variant isoform of the *INS* gene activated UPR gene expression in comparison with mock-transfected cells (Figure 5B). However, we detected statistically significant activation of regulatory elements controlling cell stress in the cells overexpressing mutant construct. The most pronounced activation was observed for the GRP78/BiP promoter element, which was almost three times higher than in the case of wild type construct. These data support our hypothesis that beta-cell mass development failure is due to inhibition of beta-cell proliferation mediated by ER stress response.

## 3. Discussion

Heterozygous dominant, compound heterozygous, and homozygous recessive mutations in the *INS* gene are among the most common causes of PNDM [30]. Dominant coding mutations are heterozygous and thought to be associated with URP, leading to ER stress and ultimately beta-cell apoptosis [31]. The recessive-acting mutations affect both alleles of the *INS* gene and have different pathogenic mechanisms, including gene deletion, abnormal transcription, lack of the translation initiation signal, and altered mRNA stability. Patients with recessive biallelic *INS* mutations exhibit a markedly different clinical phenotype with lower birth weight and earlier age at diagnosis compared with those with dominant *INS* mutations [20]. The intronic mutation c.188-31G>A in the *INS* gene is considered as a dominantly-acting mutation. The patient carrying c.188-31G>A mutation has been diagnosed with diabetes mellitus at the age of 7 months and has a normal birth weight. By 11 months, persistent decompensation of carbohydrate metabolism was observed and insulin treatment was initiated [21]. Patient’s specific iPSCs modeling provide a unique opportunity to investigate mutation impact in detail due to the recent advances in genome editing. Previously, we established iPSC line from a patient carrying c.188-31G>A the *INS* mutation. To investigate possible impact of mutation during early development, we established an isogenic model system using CRISPR/Cas9 genome editing. Despite the fact that the *INS* mutation-corrected clone was successfully obtained and expanded, some issues of CRISPR/Cas9 editing of patient’s iPSCs are worth discussing. It should be noted that intron mutations require more attention in the design of sgRNAs and ssODN. We faced the problem of ineffectiveness of restriction digest screening to reveal the corrected clones. There were no clones with disrupted BtsI restriction site generated by synonymous mutation in the ssODN among the 150 clones analyzed. This may be due to not using 3′ homologous arm of ssODN in HDR events, since the synonymous mutation is located rather distantly (about 30 bp) from the cut site or the overall homology of the chosen ssODN was lower in comparison with the homologous region of the wild type allele. Moreover, we observed biallelic deletions/insertions in some clones, which may indicate some degree of non-specificity of the selected sgRNAs. However, the cutting efficiency of sgRNAs was rather high (about 25%), which is in concordance with published data on allele-specific CRISPR/Cas9 genome editing of human iPSCs [26]. Finding an optimal trade-off of specificity and efficiency remains an important issue.

It was proposed previously that some mutations associated with PNDM manifestation are related with ectopic splicing events. Altered splicing due to homozygous mutation led to the creation of two unstable mutant transcripts and results in failure of any translated INS product in insulinoma cells [32]. The creation of an ectopic splice site in the case of heterozygous c.188-31G.A mutation was predicted in silico [20]. However, there was no direct evidence that splicing isoforms do exist in human beta-cells and it is still unknown what impact they can have on insulin-producing cells during their differentiation and development. In this study, we take advantage of somatic cell reprogramming, CRISPR/Cas9 genome editing, and direct differentiation to generate patient-specific beta-like cells that express tissue specific markers and secrete insulin. Although most secreted cells were polyhormonal as it was also observed in other studies [33,34], endogenous insulin secretion has occurred indicating *INS* expression from a wild type allele. Isogenic genetically corrected cell line did not demonstrate significant differences in insulin secretion due to intrinsic properties or variability in differentiation procedure. Therefore, the issue of splice isoform existence and its input in the *INS* gene expression was very important. For the first time, we reported the existence of splice isoform in patient’s beta-like cells in a semi-quantitative manner and evaluated the doses of the mutant and normal transcripts in the heterozygous c.188-31G>A cells. The ectopic splice site probability predicted in silico was almost 3 times higher than for the wild type splicing events, thus suggesting that splice variant could be expressed on a comparable level with the wild type transcript. Surprisingly, we found that almost two thirds of the *INS* transcripts were wild type, thus indicating that predicted probability is not very high. Although we cannot rule out the possibility that the lifetime of the isoform is shorter than the normal transcript. Similar observation in the in vitro system was recently published for another *INS* splice isoform [32]. Despite the fact that the *INS* wild type allele is expressed in a patient’s beta- cells, the impact of the mutant allele leads to the manifestation of PNDM.

Considerable advantage of iPSCs is their ability to differentiate into particular lineages. A number of protocols for PSC differentiation into pancreatic lineage were developed [8,9,13]; however, it is still difficult to reproduce them for particular cell lines or lab conditions. In our hands only one protocol [11] appeared to be reproducible, although not very efficient for all iPSC lines studied. Since populations of endocrine cells generated with the available differentiation protocols [8,9,35] are heterogeneous in terms of phenotype and maturation status, a comparative analysis is complicated even in an isogenic system. Nevertheless, for the first time, we demonstrated the difference in formation of organoids at the pancreatic progenitor stage of differentiation between patient-specific cells and isogenic wild type counterparts. Presumably, the observed differences in organoid formation were due to proliferation inhibition by the splice isoform. This is consistent with the previous results obtained in iPSC-based model of Wolfram syndrome 1, with the patient’s cells carrying mutation in the *WFS1* gene and showing incapability of forming pancreatic organoids during differentiation [10]. Moreover, we observed that cultures of mutant beta-like cells secreted lower levels of C-peptide than corrected ones. This observation was reproduced in different differentiation conditions (2D culture and 3D organoids) and at different time points (days 14 and 21). At the same time, we cannot exclude that these differences could be attributed to differentiation efficacy of independent cell lines. Single cell transcriptomic approaches may help in overcoming the problem of heterogeneity of differentiated cell populations and reveal phenotypic differences at early stages of pancreas embryonic development. 

Insulin is synthesized as preproinsulin and processed to proinsulin. Then, proinsulin is converted to insulin and C-peptide and stored as a hexameric complex in secretory granules. The existence of both mutant and wild type *INS* in the same cell leads presumably to the formation of the insoluble complexes. The intrinsic mechanisms leading to beta-cell dysfunction remain largely unknown, although previous studies implicate ER stress and aberrant UPR as key contributors to this process [12,36]. Chronic ER stress can lead to apoptosis induction [37], wherein there was no reported significant increase in beta-cell apoptosis during postnatal period [12,38,39]. The cell mass of beta-cells at this period reflects preferential proliferation of the resulting beta-cells rather than further differentiation from progenitor cells [40]. Therefore, beta-like cells differentiated from iPSCs do not putatively recapitulate postnatal beta-cells. To investigate the impact of c.188-31G>A mutation associated with PNDM on mature beta-cells, we decided to overexpress wild type and mutant *INS* transcripts in the insulinoma MIN6 cell line. Here, we showed that mutant protein significantly inhibited cell proliferation in a time dependent manner. Despite the reduction of plasmid DNA in transiently transfected cells during their cultivation due to cell division, inhibition of cell proliferation was surprisingly more prominent at day 11 than day 6. This potentially indicates that mutant protein is rather stable and forms oligomeric structures with wild type protein. This assumption is indirectly supported by the fact that we did not observe significant UPR response using conventional Western blotting. Only in a highly sensitive luciferase reporter system we detected activation of UPR associated genes. Interestingly, GRP78/BiP reporter was significantly activated by the mutant cDNA overexpression in the insulin producing cells. Recently, several studies have demonstrated that insulin can augment the UPR and protect cells against ER stress-induced apoptosis [41,42]. Therefore, GRP78/BiP could be responsible for the surveillance of beta-cells despite their underdevelopment.

In summary, we can conclude that the possible mechanism of pathogenesis of PNDM associated with c.188-31G>A heterozygous intronic mutation is decreased beta-cells proliferation due to chronic ER stress and UPR that do not involve cell apoptosis and immune response.

## 4. Materials and Methods

### 4.1. Ethics and Informed Consent

This study was approved by the Ethics Committee of the Endocrinology Research Center (Moscow, Russia), Approval number: 231/2018. Upon signed informed consent from the parents, dermal fibroblasts were derived from a skin biopsy of a 3-year-old female patient carrying *INS* c.188-31G>A mutation.

### 4.2. Cell Culture

Skin fibroblasts were reprogrammed into iPSC by transfection with non-integrating self-replicating RNA vector ReproRNA™-OKSGM (Stem Cell Technologies, Vancouver, BC, Canada) expressing OCT4, KLF-4, SOX2, GLIS1, and c-MYC, as described previously [22]. Long-term feeder-free culture of PSC lines was conducted in mTeSR1™ medium (Stem Cell Technologies) on Matrigel-coated plates (BD Biosciences, New Jersey, NJ, USA). Cells were passaged by treatment with the Versene solution (PanEco, Moscow, Russia) every 4–6 days in medium supplemented with the 10 μm ROCK Inhibitor Y-27632 (Stem Cell Technologies) one day after passaging. ESC H9 cell line was used as a gold standard control for the differentiation into pancreatic lineage and was cultivated as described above. The H9 cell line was provided by the WiCell Research Institute, Inc., Madison, WI, USA. MIN6 mouse insulinoma cells were cultured in Dulbecco’s modified Eagle’s medium (25 mM glucose) supplemented with 15% fetal bovine serum, GlutaMAX, 10 mM HEPES, 1% penicillin/streptomycin, and 50 μM β-mercaptoethanol. Cells were maintained in the atmosphere of 5% CO_2_. HEK293 cell line was cultured in DMEM (25 mM glucose) supplemented with 10% fetal bovine serum, GlutaMAX, and 1% penicillin/streptomycin. MIN6 cell line was cultured in DMEM (25 mM glucose) supplemented with 10% fetal bovine serum, GlutaMAX, sodium pyruvate, and 1% penicillin/streptomycin.

### 4.3. In Vitro Differentiation of PSC Lines to Beta-Like Pancreatic Cells

For in vitro pancreatic endocrine cell differentiation, we used a previously described protocol [11] with minor modifications (Appendix A). Briefly, 80–90% confluence cells were passaged as single cells with Versene solution (Paneco) and 1 million dissociated cells were seeded onto 10 cm^2^ Matrigel-coated plates in mTeSR1 medium (Stem Cell Technologies) containing 5 μM Y26732. After 24 h, beta-cell differentiation was started by replacing mTeSR1 medium with STEMDiff Endoderm Differentiation Kit (Stem Cell Technologies). DE differentiation lasted 4 days, and cells were analyzed at day 5 of differentiation for the DE markers. At day 5, media was changed to RPMI-1640 supplemented with 1 × B27 as well as 50 ng/mL FGF7 (fibroblast growth factor) for 2 days. DMEM with high glucose supplemented with pyruvate ×1 B27 and 2 μM all-trans-retinoic acid, 0.25 μM cyclopamine, and 250 nM LDN193189 containing medium was used for additional three days. Media was changed every 36 h. For the next 3 days, DMEM with high glucose was supplemented with pyruvate ×1 B27 and 50 ng/mL EGF-containing medium for another 3 days. Upon reaching the PP stage, cells were dissociated into single cells, and seeded in high density (5 × 104 cells per 1 well of 6-well plate) in media supplemented with Y-27632 5 µM and cultured on low-attachment surfaces (Costar Corning, Corning, NY, USA) for 1 more week. Alternatively, cells were intact and only the media was changed from Stage 4 for Stage 5 media. Stage 5 medium contains cyclopamine 0.25 µM, T3 1 µM, Alk5i 10 µM, zinc-sulfate 10 µM, and heparin 10 µg/mL. The list of reagents used in the differentiation is present in Appendix A.

### 4.4. sgRNA Design and Cloning

The online tool https://benchling.com/ (accessed on 14 January 2019) was used to design sgRNA. Five extra bases were added to the 3′ end of each single strand sgRNA oligonucleotide to enable direct cloning into GeneArt^®^ CRISPR Nuclease Vector (Thermo Fisher Scientific, Waltham, MA, USA) under the control of U6 promoter (Appendix A). The corresponding single-strand oligonucleotides were synthesized by Evrogen (Moscow, Russia). The oligonucleotides were annealed and ligated with GeneArt^®^ CRISPR Nuclease Vector. Several ampicillin-resistant colonies obtained after transformation of E. coli were analyzed via plasmid DNA isolation. To verify correct sgRNA insertion into the GeneArt^®^ CRISPR Nuclease Vector, plasmid DNA was sequenced using the U6 promoter forward primer 5′-GGACTATCATATGCTTACCG-3′.

### 4.5. Cloning of the INS Gene and Plasmid Construction

Fragments of the *INS* gene were amplified from patient’s genomic DNA by PCR with primers Ins-CDS-RI-F and R1InsEx3-R. PCR products encompassing 1187 bp with genome coordinates chr11:2159800-2160986 (GRCh38/hg38 assembly, UCSC Genome Browser) were purified by 1% agarose gel electrophoresis, digested with EcoRI restriction endonuclease, and cloned in pcDNA3.1(+) vector (Thermo Fisher Scientific). Recombinant clones containing wild type and mutant allele were selected by colony PCR and sequencing. HEK293 cells were transfected with wild type and mutant allele plasmids, mRNA was isolated and reverse transcribed using ImProm-IITM Reverse Transcription System (Promega Corp., Madison, WI, USA) with oligo(dT)15 Primer. The *INS* cDNA fragments were amplified and cloned in pcDNA3.1(+) vector. Plasmids pcDNA3.1-preproIns WT and pcDNA3.1-preproIns SNP were isolated, verified by sequencing, and used in transfection experiments. To enable transfected cell sorting, GFP was introduced at 3′ end of the cDNAs through the IRES element in the constructs mentioned above. Sequence orientation and correctness were confirmed by sequencing. To confirm the existence of splice mRNA isoform in beta-like cells, we isolated RNA from isogenic cell lines and H9 control cell line, converted it to cDNA, amplified the region of interest of the *INS* gene, and cloned into pcDNA3.1(+) vector. Individual E.coli clones were sequenced. All primer sequences are shown in Appendix A.

The human genomic fragment containing Grp78 promoter was amplified with primers GRP78-LucF -3′ and Grp78-LucR and cloned using KpnI and XhoI restriction endonucleases into pGL3-basic vector (Promega Corp.). To construct plasmid, pGL3-UPRE containing five copies of UPR was assembled in two steps using two pairs of oligonucleotides: UPRE-1-For and UPRE-1-Rev, UPRE-2-For and UPRE-2-Rev. Then, DNA duplex was inserted into pGL3-promoter vector (Promega Corp.) using KpnI and XhoI restriction endonucleases. pGL3-ERSE vector containing ER stress response element was prepared by annealing oligonucleotides ERSE-For and ERSE-Rev and inserted into pGL3-promoter vector digested with KpnI and XhoI restriction endonucleases. All primer sequences are shown in Appendix A.

### 4.6. Cells Transfection and iPSC Clone Selection

Patient-derived iPSCs were transfected by reverse lipofection with TransIT^®^-LT1 Transfection Reagent (Mirus Bio, Madison, WI, USA). The transfection efficiency was monitored by determining OFP expression via fluorescence microscopy. At 48 h post-transfection, OFP-expressing transfected cells were analyzed and collected by fluorescence-activated cell sorting (FACS). Briefly, cells were treated with EDTA solution, resuspended in mTeSR1 medium, and filtered through a 70 μm cell strainer. Next, cells were loaded into a Sony MA900 sorter (Sony Biotechnology Inc., Tokyo, Japan). Detection of the target population of OFP-positive cells was conducted using a 488 nm laser in both the PE and FITC channels to separate the fluorescence signal from the autofluorescence background. Cells were sorted in 6-well plates, 1000–2000 cells per well.

### 4.7. DNA, RNA Extraction, Sanger Sequencing

The genomic DNA was extracted with Extract DNA Blood Kit (Evrogen, Moscow, Russia). The region of interest in the *INS* gene was amplified via PCR using primers presented in Appendix A. To confirm the presence of mutation, the PCR products were analyzed by Sanger sequencing with primers presented in Appendix A. PCR was performed via ScreenMix PCR amplification mix (Evrogen). RNA was extracted via Extract RNA kit (Evrogen).

### 4.8. Immunocytochemistry

The pancreatic markers were assessed by immunofluorescence staining. Cells were fixed in 4% methanol-free paraformaldehyde (Thermo Fisher Scientific), washed with phosphate buffer saline (PBS) (Thermo Fisher Scientific), permeabilized with 0.5% Triton-X100, and blocked with 2.5% bovine serum albumin (BSA) (Paneco) for 30 min. After overnight incubation at 4 °C with primary antibodies specific to Sox17, CXCR4, FOXA2, PDX1, NKX6.1, insulin, C-peptide, glucagon, cells were washed and incubated with secondary antibodies for 1 h at room temperature. The antibodies are summarized in Appendix A. DAPI (Thermo Fisher Scientific) was used for nuclear counterstaining. The images were obtained by fluorescence microscopy using inverted microscope ZEISS Axio Observer with the Colibri 5 LED (Carl Zeiss, Göttingen, Germany) using ZEN software (Carl Zeiss).

### 4.9. Cell Proliferation Measurement

To compare growth rates of MIN6 cells under influence of the wild type and mutant *INS* cDNA expressing constructs, 2 × 10^6^ cells were seeded into 6-well plates and transfected with GFP tagged wild type and mutant constructs, according to the manufacturer’s recommendations. pMAX-GFP (Lonza, Basel, Switzerland) was used as a control. Transfection efficiency was routinely 40%. After 2 days, transfection cells were sorted using Sony MA900 sorter (Sony Biotechnology Inc.) and plated for additional 4 days to grow. For proliferation measurement, cells were seeded in two replicates into wells of a 16-well xCELLigence (Agilent, Santa Clara, CA, USA) plate, according to the manufacturer’s recommendations. As the xCELLigence in real time measures the net adhesion of cells to high-density gold electrodes, we analyzed cell index curves. As a baseline background, we used the cell index level from the culture medium with no cells. We measured the cell growth before a plateau phase of proliferation. The curves represent the mean cell index value from minimum 2 wells ± SD. To compare the growth rate, the slope of the linear region of the cell index curve was calculated using the internal xCELLigence software. Transfection, sorting, and proliferation measurements were repeated twice.

### 4.10. Reporter Assays

MIN6 cells were seeded in 24-well culture plates. After 24 h, cells were transiently transfected using TurboFect (Thermo Fisher Scientific), according to the manufacturer’s recommendations. Cells were transfected with plasmids encoding human pcDNA-Ins WT or pcDNA-Ins SNP and pGL3-Grp78 (BiP), pGL3-ERSE or pGL3-UPRE luciferase reporters. Co-transfection with the plasmid encoding β-galactosidase (pcDNA3.1-lacZ) was used as a control for transfection efficiency; pcDNA3.1(+) plasmid was used as a control for basal transcriptional activity. At 48 h post-transfection, cells were washed with PBS and then lysed with Glo Lysis Buffer (Promega Corp.). Supernatants were collected for measurement of firefly luciferase activity using ONE-Glo Ex Luciferase Assay System (Promega Corp.). In all experiments, luciferase activity was controlled by activity of β-galactosidase, which was measured by incubation of cell lysates with 100 mM sodium phosphate buffer pH 7.4 containing 2.92 mM ONPG (o-nitrophenyl- β-D-galactopyranoside), 1 mM MgCl2, and 45 mM β-mercaptoethanol. The β-galactosidase activity was quantified at 420 nm using a microplate reader Tecan SPARK (Tecan Group Ltd., Männedorf, Switzerland). Each experiment was performed independently three times in triplicate. Results are shown as means ± SD.

### 4.11. ELISA

HEK293 and MIN6 cells were seeded in a 24-well plate the day before transfection. Then, cells were transiently transfected with 500 ng of plasmids encoding pcDNA3.1-Ins WT or pcDNA3.1-Ins SNP and pcDNA3.1-LacZ using TurboFect, according to the manufacturer’s recommendations. At 48 h post-transfection, cells were washed with PBS and lysed with Passive Lysis buffer (Promega Corp.). Supernatants of HEK293 cells were collected and then used to measure intracellular proinsulin content using Human Intact Proinsulin ELISA kit (Biovendor-Laboratorni medicina a.s., Brno, Czech Republic) following instructions. MIN6 cell extracts were used to evaluate levels of intracellular insulin or C-peptide using AccuBind ELISA Microwells kits (Monobind Inc., Lake Forest, CA, USA). Results were normalized to β-galactosidase activity to control transfection efficiency. Each experiment was performed independently three times in triplicate. Data are presented as means ± SD.

### 4.12. Statistical Analysis

Results were analyzed by one-way ANOVA followed by Dunnett’s post-hoc test for multiple comparisons using GraphPad Prism 6. A *p*-value of 0.05 was considered significant.

## Figures and Tables

**Figure 1 ijms-23-08824-f001:**
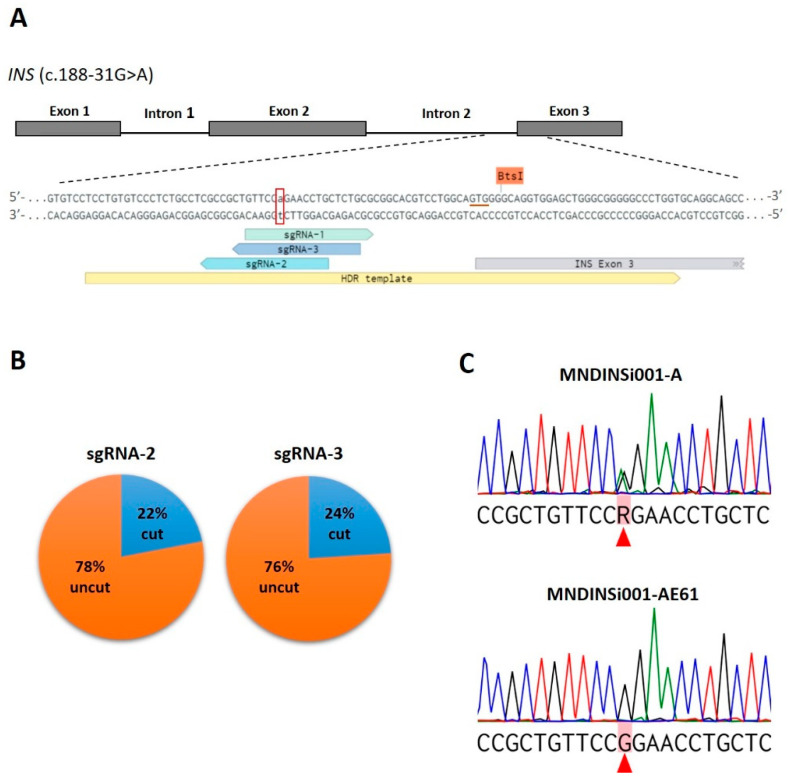
CRISPR/Cas9-mediated correction of a heterozygous mutation c.188-31G>A in the *INS* gene in patient-specific iPSCs. (**A**) Schematic representation of the mutated region of *INS* with designed allele-specific sgRNAs (sgRNA-1, -2, -3) and ssODN used as HDR template. The mutated nucleotide (adenosine) is indicated by lowercase letter and labeled with a red open box. The underlined triplet GTG (in red color) indicates a position of synonymous mutation (GTC) in the ssODN for disrupting BtsI restriction site. (**B**) Pie charts show the cutting efficiency of sgRNAs observed in CRISPR/Cas9-edited patient-specific iPSCs. (**C**) Sequencing chromatograms of patient-specific MNDINSi001-A cells with the heterozygous mutated alleles of *INS* and successfully edited clone MNDINSi001-AE61 with wild type alleles. Red triangles indicate heterozygous mutated and the corresponding corrected nucleotides. R = G or A according to the IUPAC nucleotide code.

**Figure 2 ijms-23-08824-f002:**
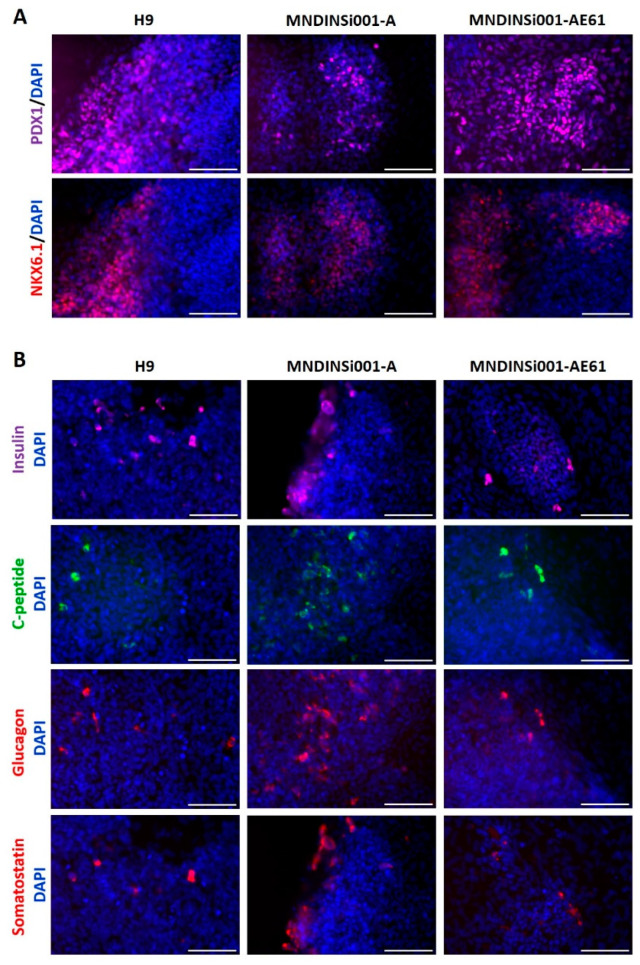
The potential of isogenic iPSC lines to differentiate into pancreatic lineage. (**A**) Immunocytochemistry for pancreatic progenitor markers PDX1 and NKX6.1 in H9 (gold-standard), MNDINSi001-A (mutant), and MNDINSi001-AE61 (CRISPR/Cas9-corrected) cell lines at day 14 of differentiation. (**B**) Immunocytochemistry for insulin and C-peptide (beta-cell hormones), glucagon (alpha cell hormone), and somatostatin (delta-cell hormone) in H9 (gold-standard), MNDINSi001-A (mutant), and MNDINSi001-AE61 (corrected) cell lines at day 14 of differentiation. Scale bars, 100 µm.

**Figure 3 ijms-23-08824-f003:**
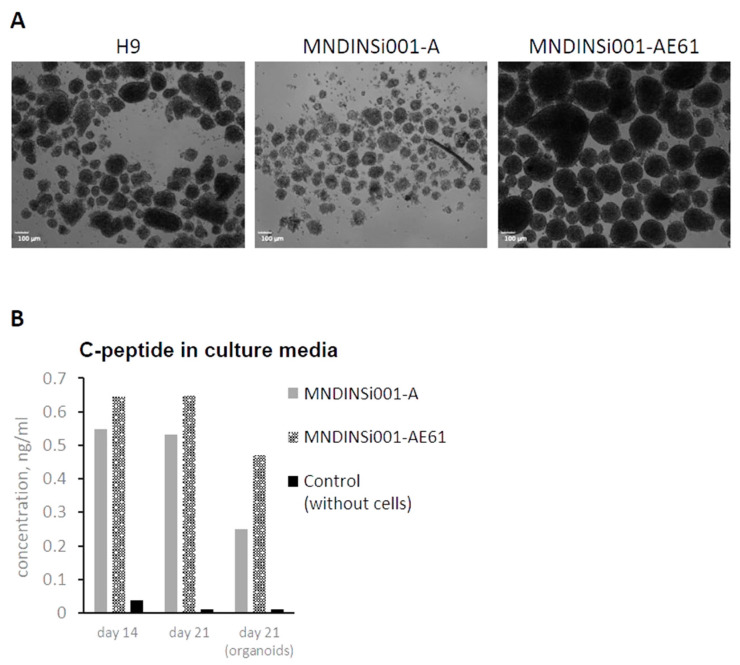
Organoid formation and C-peptide secretion at late stages of pancreatic differentiation in an isogenic system. (**A**) Abnormal organoids generated from MNDINSi001-A (mutant) cells compared with ones generated from isogenic control MNDINSi001-AE61 (CRISPR/Cas9-corrected) cells at day 21 of pancreatic differentiation. H9 ESC line was used as a gold standard. (**B**) MNDINSi001-A (mutant) and MNDINSi001-AE61 (CRISPR/Cas9-corrected) beta-like cells secreted C-peptide into culture media at days 14 and 21. Data are shown for the representative differentiation experiment.

**Figure 4 ijms-23-08824-f004:**
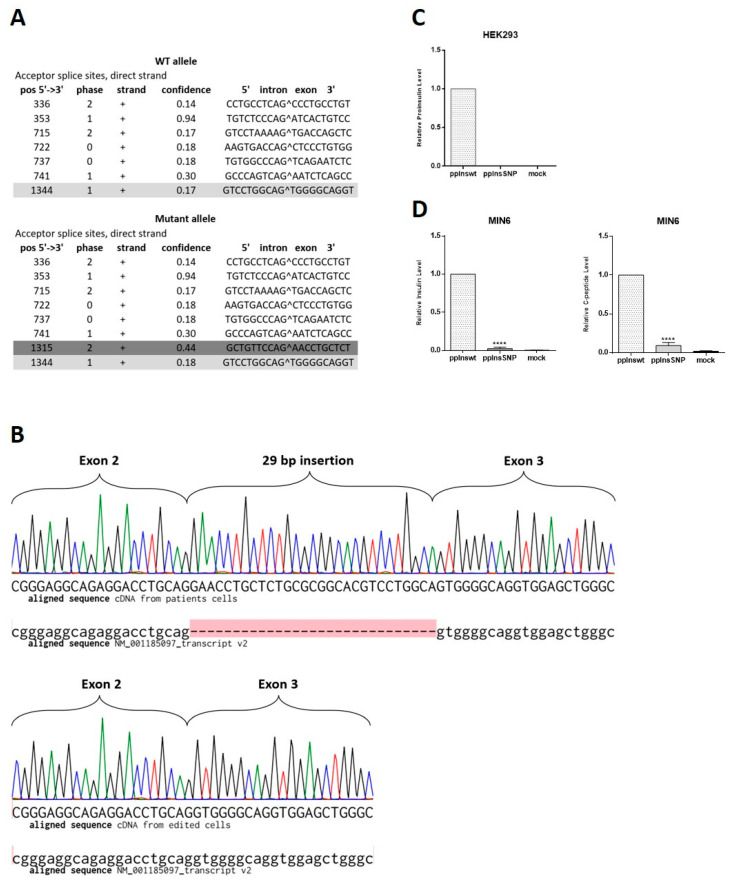
The c.188-31G>A *INS* splice variant detection and analysis of its influence on insulin expression. (**A**) Splice variant prediction. Dark grey color shows normal splice site between exon 2 and 3 with the confidence 0.17. Bright grey color marks the new ectopic splice site with the higher confidence (0.44) in comparison with the normal site (0.18). (**B**) Sequencing chromatograms of the region of *INS* cDNA obtained from mRNA of iPSC-derived beta-like cells aligned to *H. sapiens* insulin, transcript variant 2, mRNA (NM_001185097.2). Upper panel shows mutant allele in cDNA library from patient’s cells, where the red highlight indicates 29 bp insertion that is absent in NM_001185097.2, lower panel shows wild type allele in cDNA library from CRISPR/Cas9-edited cells. (**C**) HEK293 cells were transiently transfected with pcDNA3.1-preproIns wild type or SNP. The data are normalized to activity of beta-galactosidase. Intracellular level of proinsulin is shown as means ± SD (*n* = 3). (**D**) MIN6 insulinoma cells were transiently transfected with pcDNA3.1-preproIns wild type or SNP. Intracellular levels of synthesized insulin (left) and C-peptide (right) are shown as means ± SD (*n* = 3); **** *p* < 0.0001.

**Figure 5 ijms-23-08824-f005:**
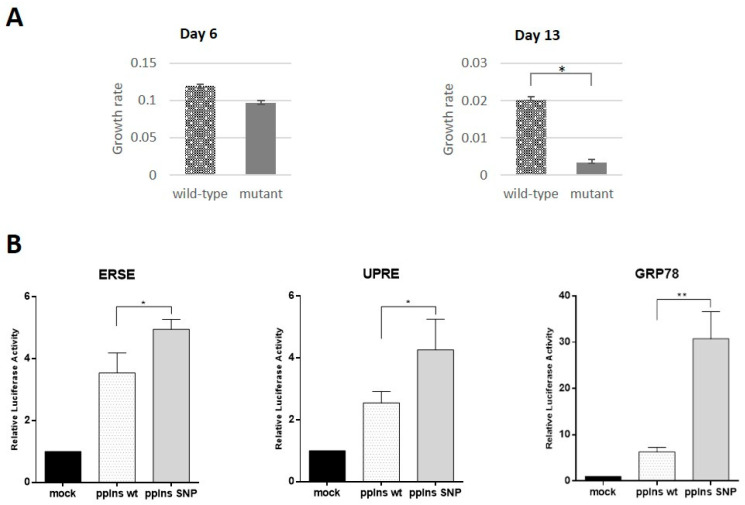
The *INS* splice isoform inhibits growth rate and activates endoplasmic reticulum stress response in MIN6 insulinoma cells. (**A**) Proliferation rate of MIN6 cells transfected with the mutant and wild type *INS* cDNA sequences at days 6 and 13 after transfection. (**B**) Activation of luciferase gene under control of ER stress response element (ERSE) (left panel), unfolded protein response element (UPRE) (middle panel) or GRP78/BiP promoter (right panel) in MIN6 cells in the presence of phCMV-preproIns wild type and SNP are shown. The data are presented as means ± SD (*n* = 3), * *p* < 0.05, ** *p* < 0.01. All results are normalized to the activity of β-galactosidase.

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
