# Peer review of "Aberrant Splicing of INS Impairs Beta-Cell Differentiation and Proliferation by ER Stress in the Isogenic iPSC Model of Neonatal Diabetes"

_ijms, 2022, doi:10.3390/ijms23158824_

Round 1
Reviewer 1 Report
This manuscript “Aberrant splicing of INS impairs beta-cell differentiation and 2 proliferation by ER stress in the isogenic iPSC model of neonatal diabetes”. prepared by author Alexandra V. Panova et al. Overall it is a very interesting paper, but appropriate wording is needed and several issues need to be addressed as follows: Major Issues: 1. The patient with neonatal diabetes mellitus carrying the INS mutation in the 2nd intron (с.188-31G>A), this is a rare disease. Please included more information about this patient and whether this mutant had a pattern of family inheritance is not mentioned. 2. For this mutation (), after transcription, what is amino acid sequence, dose this mutation cause the change a folded protein structure and function? 3. In Figure 3, when comparing the abnormal organoids generated from MNDINSi001-A (mutant) cells and isogenic control MNDINSi001-AE61 (CRISPR/Cas9-corrected) cells at day 21 of pancreatic differentiation,the picture of H9 and MNDINSi001-AE61 are exactly the same. Please double check this issue. 4. Line 183, author mentioned they measured insulin and C-peptide levels in the culture media. However, in the Figure 3B, only the C-peptide data was showed, how many samples in each group was not mentioned. Furthermore, whether there are significant difference expression levels of insulin and C-peptide between MNDINSi001-A 195 (mutant) and MNDINSi001-AE61 (CRISPR/Cas9-corrected) beta-like cells ? 5.For immunocytochemistry part, what is the concentration that was used for the primary antibodies ? 6. Line 500, 2x106 should be 2x106.
Author Response
RESPONSE TO REVIEWER 1:
We are grateful for your interest in our manuscript and for valuable advices. We appreciate all suggestions which improve the manuscript. Herein we submit the revised version of the manuscript addressing your comments.
REVIEWER 1.
Major Issues:
- The patient with neonatal diabetes mellitus carrying the INS mutation in the 2nd intron (с.188-31G>A), this is a rare disease. Please included more information about this patient and whether this mutant had a pattern of family inheritance is not mentioned.
We are grateful for your comment. We added the following sentences in the Results section (please see lines 90-93 highlighted in yellow):
“We generated an iPSC line from a patient with PNDM carrying an intronic mutation с.188-31G>A in the INS gene. Diabetes mellitus was diagnosed at the age of 7 months during a scheduled examination for glucosuria. The disease first manifested as hyperglycemia up to >14 mmol/L in the absence of beta-cell autoantibodies. No family anamnesis was found [21].”
- For this mutation (), after transcription, what is amino acid sequence, dose this mutation cause the change a folded protein structure and function?
Thank you for your question. As it was discussed in the previously published papers (Garin et al, 2012, 10.1371/journal.pone.0029205, Panova et al, 2020, 10.1016/j.scr.2020.101929), the substitution с.188-31G>A in the INS gene is intronic and supposed to create an ectopic splice site, which results in a longer, abnormal transcript. The 29 bp insertion alters the reading frame in the mutant transcript and the extended protein was predicted to be misfolded in the endoplasmic reticulum followed by beta-cell apoptosis (Garin et al 2012). Therefore, we expanded the Introduction section as follows (please see lines 74-77 highlighted in yellow):
“To study the role of an intronic mutation we established iPSC lines from a patient carrying the INS mutation in the 2nd intron (с.188-31G>A). The previous study suggested that this substitution creates an ectopic splice site, which results in a longer, abnormal transcript and presumably leads to the proinsulin misfolding followed by ER stress and beta-cell death [20]. “
- In Figure 3, when comparing the abnormal organoids generated from MNDINSi001-A (mutant) cells and isogenic control MNDINSi001-AE61 (CRISPR/Cas9-corrected) cells at day 21 of pancreatic differentiation,the picture of H9 and MNDINSi001-AE61 are exactly the same. Please double check this issue.
Thank you for your remark. We are deeply sorry this confusion. We replaced the picture of H9 organoids. Please, find the new data on Figure 3A.
- Line 183, author mentioned they measured insulin and C-peptide levels in the culture media. However, in the Figure 3B, only the C-peptide data was showed, how many samples in each group was not mentioned. Furthermore, whether there are significant difference expression levels of insulin and C-peptide between MNDINSi001-A 195 (mutant) and MNDINSi001-AE61 (CRISPR/Cas9-corrected) beta-like cells ?
Thank you for your questions. Indeed, we measured both insulin and C-peptide levels in the culture media of MNDINSi001-A 195 (mutant) and MNDINSi001-AE61 (CRISPR/Cas9-corrected) beta-like cells. However, we found out that results may be confusing due to the presence of insulin in the differentiation medium, therefore, we focused on C-peptide measurements.
Therefore, we corrected the manuscript according to your remark (please see lines 193-199 highlighted in yellow):
“We measured C-peptide level, a proinsulin-cleavage product that is co-secreted with insulin by pancreatic beta-cells, in culture media at days 14 and 21 using planar cell culture and organoids. In all culture conditions and time points we detected C-peptide secretion (fresh cell culture medium served as a control), meaning that differentiated cells were capable to secrete beta-cell hormone during early differentiation events (Figure 3B). “
We would like to admit that we performed several rounds of patient’s iPSC differentiation into beta cells including 3 different protocols (Rezania et al, 2014 10.1038/nbt.3033, Ma et al, 2018 10.1016/j.stemcr.2018.11.006, and commercial kit from STEMCELL Technologies). C-peptide level was assessed just to demonstrate by different methods that differentiated cells were capable of beta-cell hormone secretion. Figure 3B shows the data from one differentiation experiment. This is an example of a qualitative experiment. As we expected, we did not observe significant differences in the expression levels of C-peptide between MNDINSi001-A 195 (mutant) and MNDINSi001-AE61 (CRISPR/Cas9-corrected) beta-like cells.
We updated Figure 3B legend (please see lines 207-208 highlighted in yellow):
“Data are shown for the representative differentiation experiment.”
We also expanded the Discussion section according to your comments (please see lines 382, 388-392 highlighted in yellow):
Nevertheless, we for the first time demonstrated the difference in formation of organoids at the pancreatic progenitor stage of differentiation between patient-specific cells and isogenic wild-type counterparts. Presumably the observed differences in organoid formation were due to proliferation inhibition by the splice isoform. This is consistent with the previous results obtained in iPSC-based model of Wolfram syndrome 1 patient’s cells carrying mutation in the WFS1 gene showed incapability to form pancreatic organoids during differentiation [10]. We also observed that cultures of mutant beta-like cells secreted lower levels of C-peptide than corrected ones. This observation was reproduced in different differentiation conditions (2D culture and 3D organoids) and at different time points (day 14 and day 21). At the same time we cannot exclude that these differences could be attributed to differentiation efficacy of independent cell lines.
5.For immunocytochemistry part, what is the concentration that was used for the primary antibodies ?
Thank you for your suggestion. We added the information on concentration of primary antibodies used in the study in Table S8. Please find the updated table in the Supplementary file.
- Line 500, 2x106 should be 2x106.
Thank you for your note. We changed the formatting to superscript symbol (please see lines 537 highlighted in yellow).
Reviewer 2 Report
This manuscript focuses on understanding the effect of a human INS gene mutation on beta-cell development and function. The mutation occurs in an intron, resulting in aberrant mRNA splicing. The work aimed at disease modeling using beta-like cells differentiated from patient skin fibroblast-derived iPS cells. Mutant beta-like cells were compared to isogenic cells, in which the mutation was corrected at the iPS stage using CRISPR-Cas9.
Although INS gene mutations represent a rare cause of type 1 diabetes, this ambitious approach could shed light on important aspects of beta-cell development and function. However, the characterization of the differentiated beta-like cells is partial, and the authors opt instead to perform some of the studies in a murine transformed beta-cell line, MIN6, transfected with mutant and corrected human INS genes.
The Results and Discussion leave multiple issues unaddressed or not clearly explained:
1. Does the mutant mRNA produce protein? Some of the results suggest it does not, but the Discussion (lines 378-9) states that “mutant protein is rather stable”.
2. The patients are heterozygous for the mutation. Is it dominant? If so, why?
3. Individual “clones” of beta-like cells derived from the mutant iPS line differ in expression of the mutant and WT mRNA, with each clone expressing only one of them. It is unclear what the authors mean by “clones”, as iPS cell lines are usually clonal. It is not explained why would individual cells express only one type of mRNA, when all contain both alleles.
4. The failure of mutant beta-like cells to efficiently form organoids (Fig. 3A) is not explained. How can lack of insulin production in some cells affect that?
5. The mutant gene reduces growth of MIN6 cells (Fig. 5A). As in point 4 above, it is unclear how lack of insulin production from the mutant gene can cause this effect, as the cells continue to express mouse insulin. Similarly, why should it cause ER stress?
6. Why is ER stress in MIN6 cells (Fig. 5B) assessed by reporter gene expression rather than more direct methods?
7. A major shortcoming of the study is that it is based on a single iPS line. Confidence in the results would be greatly increased by showing reproducibility with multiple patient-derived lines.
Most of these issues could be explained if the mutant gene produces an abnormal INS protein, which complexes with WT protein and prevents its processing or transport, thus leading to ER stress and cell death. However, this possibility is not examined experimentally in this work. The authors should make use of the abundant iPS-derived beta-like cells to address this possibility at the protein and ultrastructural levels.
Author Response
REVIEWER 2.
This manuscript focuses on understanding the effect of a human INS gene mutation on beta-cell development and function. The mutation occurs in an intron, resulting in aberrant mRNA splicing. The work aimed at disease modeling using beta-like cells differentiated from patient skin fibroblast-derived iPS cells. Mutant beta-like cells were compared to isogenic cells, in which the mutation was corrected at the iPS stage using CRISPR-Cas9.
Although INS gene mutations represent a rare cause of type 1 diabetes, this ambitious approach could shed light on important aspects of beta-cell development and function. However, the characterization of the differentiated beta-like cells is partial, and the authors opt instead to perform some of the studies in a murine transformed beta-cell line, MIN6, transfected with mutant and corrected human INS genes.
The Results and Discussion leave multiple issues unaddressed or not clearly explained:
RESPONSE TO REVIEWER 2
We would like to thank you for careful reading of our manuscript and valuable remarks. We appreciate your suggestions that make the manuscript more clear for the readers. Herein we submit the revised version of the manuscript addressing your comments.
- Does the mutant mRNA produce protein? Some of the results suggest it does not, but the Discussion (lines 378-9) states that “mutant protein is rather stable”.
Thank you for your question. As it was discussed previously (Garin et al, 2012, 10.1371/journal.pone.0029205), the substitution с.188-31G>A in the INS gene is intronic and supposed to create an ectopic splice site, which results in a longer, abnormal transcript. In our study the existence of the in silico predicted isofom has been experimentally demonstrated for the first time. Since the frame shift does not generate a premature termination codon the new stop codon would be located within the 3′ UTR. This fact was also confirmed in our study by using in-frame fused GFP for cell sorting (see Fig 5 and Material and methods section). Presumably, the mutant protein would normally translocate into the ER as the signal peptide is intact. In silico 3D structure studies of its amino acid sequence suggest that mutant protein would fail to fold properly (Garin et al 2012). We were unable to detect any cross-homology with proinsulin, C-peptide, or insulin using ELISA (Fig. 4C,D) in HEK293 and MIN6 cells. Indeed, we do not have any direct evidence on the stability of the mutant protein, that is why in the Discussion section we wrote (please see lines 412 highlighted in yellow): “This potentially indicates that mutant protein is rather stable and forms oligomeric structures with wild type protein.” speculating on our proliferation experiments.
- The patients are heterozygous for the mutation. Is it dominant? If so, why?
Thank you for your question. Yes, the mutation с.188-31G>A in the INS gene is considered as a dominantly-acting mutation. The recessive-acting mutations affect both alleles of the INS gene and are characterized by a markedly different clinical phenotype in patients with lower birth weight and earlier age at diagnosis compared to dominant INS mutations. [10.1073/pnas.0910533107]. The patient carrying с.188-31G>A mutation has been diagnosed at the age of 7 months and has a normal birth weight. By 11 months persistent decompensation of carbohydrate metabolism was observed and insulin treatment was initiated (Tikhonovich et al 2022 10.14341/DM12737).
To address your question we expanded the Discussion section as follows (please see lines 313, 314-323 highlighted in yellow):
Dominant coding mutations are heterozygous and thought to be associated with URP, leading to ER stress and ultimately beta cell apoptosis [31]. The recessive-acting mutations affect both alleles of the INS gene and have different pathogenic mechanisms including gene deletion, abnormal transcription, lack of the translation initiation signal, and altered mRNA stability. Patients with recessive biallelic INS mutations exhibit a markedly different clinical phenotype with lower birth weight and earlier age at diagnosis compared to those with dominant INS mutations [20]. The intronic mutation с.188-31G>A in the INS gene is considered as a dominantly-acting mutation. The patient carrying с.188-31G>A mutation has been diagnosed for diabetes mellitus at the age of 7 months and has a normal birth weight. By 11 months persistent decompensation of carbohydrate metabolism was observed and insulin treatment was initiated [21].”
and (please see lines 350-353 highlighted in yellow):
“Altered splicing due to homozygous mutation led to creation of two unstable mutant transcripts and results in failure of any translated INS product in insulinoma cells [32]. The creation of an ectopic splice site in the case of heterozygous c.188-31G.A mutation was predicted in silico [20].”
- Individual “clones” of beta-like cells derived from the mutant iPS line differ in expression of the mutant and WT mRNA, with each clone expressing only one of them. It is unclear what the authors mean by “clones”, as iPS cell lines are usually clonal. It is not explained why would individual cells express only one type of mRNA, when all contain both alleles.
Thank you for your question. Indeed, there is a confusing terminology in mammalian cells clone selection and E.coli clones analysis. We used the term “clones” in this context meaning standard cloning technique. Regarding iPSC clone selection, this procedure was performed during generation of CRISPR/Cas9-edited cell lines. To make it more clear for the Readers, we highlighted this in the title of 4.6 section of Materials and Methods (please see lines 507 highlighted in yellow) “4.6. Cells transfection and iPSC clone selection”.
To demonstrate the existence of mRNA splice isoform in beta-like cells we isolated RNA from iPSC differentiated into beta-like cells and converted it to cDNA. The region of interest was amplified and cloned in the plasmid vector. Plasmids from individual E.coli clones were sequenced. To make it more clear for the Readers we introduced the following sentences in the 4.5 section of Materials and Methods (please see lines 493-496 highlighted in yellow): “To confirm the existence of splice mRNA isoform in beta-like cells we isolated RNA from isogenic cell lines and H9 control cell line, converted it to cDNA, amplified the region of interest of the INS gene, and cloned into pcDNA3.1(+) vector. Individual E.coli clones were sequenced.”
- The failure of mutant beta-like cells to efficiently form organoids (Fig. 3A) is not explained. How can lack of insulin production in some cells affect that?
Thank you for your interest. Actually, we could not say that we observed the lack of insulin production, although different levels of secretion could be expected. Using MIN6 cells we demonstrated that expression of splice isoform statistically significantly inhibited beta cells proliferation along with insulin production. Thus we can speculate that proliferation of even polyhormonal cells that form organoids is also inhibited. To make it more clear for the Readers we added the following sentence to the Discussion section (please see lines 384-385 highlighted in yellow): “Presumably the observed differences in organoid formation were due to proliferation inhibition by the splice isoform. “
- The mutant gene reduces growth of MIN6 cells (Fig. 5A). As in point 4 above, it is unclear how lack of insulin production from the mutant gene can cause this effect, as the cells continue to express mouse insulin. Similarly, why should it cause ER stress?
Thank you for your question. As it was mentioned above the intronic mutation in the INS gene acts dominantly despite the presence of the wild-type allele. There is no lack of insulin production in patient’s beta-like cells (Fig. 2B) or in MIN6 insulinoma cells overexpressing mutant cDNA. Upon processing insulin stored as hexameric complex in granules before secretion from beta cells. The existence of both mutant and wild-type INS in the same cell leads presumably to the formation of the insoluble complexes of the unknown structure, which results in ER stress and inhibition of cell proliferation, as it was demonstrated in our manuscript. To make it more clear for the Readers we introduced the following sentences in the Discussion section (please see lines 396-399 highlighted in yellow):
“Insulin is synthesized as preproinsulin and processed to proinsulin. Proinsulin is then converted to insulin and C-peptide and stored as hexameric complex in secretory granules. The existence of both mutant and wild-type INS in the same cell leads presumably to the formation of the insoluble complexes.”
- Why is ER stress in MIN6 cells (Fig. 5B) assessed by reporter gene expression rather than more direct methods?
Thank you for your question. Indeed, we initially tried to use conventional and simple methods to analyze ER stress in MIN6 cells. As we indicated in the 2.5 subsection of the Results section “We used ER Stress Antibody Sampler Kit, that contains a pool of molecular chaperone proteins including calnexin, GRP78/BiP, PDI, CHOP and others to analyze protein extracts from beta-like cells differentiated from isogenic iPSC lines and MIN6 cells transfected with wild-type and mutant cDNAs. We did not find any differences between wild-type and mutant alleles by Western blotting (data not shown).“ The absolute number of differentiated beta-like cells or transfected MIN6 cells perhaps did not allow us to detect the ER stress markers by conventional approach, thus we decided to use a more sensitive and quantitative method based on the luciferase gene reporter system, although it was an effort and time consuming approach.
- A major shortcoming of the study is that it is based on a single iPS line. Confidence in the results would be greatly increased by showing reproducibility with multiple patient-derived lines.
Thank you for your meaningful remark. We agree that usage of multiple patient-derived lines expands the possibilities for research and increases reliability of the results. However, this is often unavailable. The pathogenic variant p.188–31G>A in the intron of the INS gene is a rare mutation. In Russia, it has been identified for the first time by our group (Tikhonovich et al., 2021). Regarding phenotype-genotype relationships this clinical case is quite special with no clear understanding of pathogenetic mechanisms. We for the first time demonstrated the existence of p.188–31G>A splice isoform in patient-derived beta cells and its influence on beta-cell differentiation and organoid formation that can be useful and interesting for the diabetes research community.
Most of these issues could be explained if the mutant gene produces an abnormal INS protein, which complexes with WT protein and prevents its processing or transport, thus leading to ER stress and cell death. However, this possibility is not examined experimentally in this work. The authors should make use of the abundant iPS-derived beta-like cells to address this possibility at the protein and ultrastructural levels.
We absolutely agree with your valuable remark. Indeed, mutant protein undoubtedly complexes with wild-type protein and prevents some stages of insulin processing inhibiting beta-cell proliferation. In our present study we demonstrated that this occurs during early development (organoid formation) and in mature beta cells (MIN6 insulinoma cells). Currently we are planning to study mutant and WT proteins interaction on an ultrastructural level using MIN6 cells that stably transfected with the fluorescently tagged constructs. We hope that this model will help us to find therapeutic targets to treat newborns. Along with it we are planning to improve iPSC differentiation procedure that will enable us to work with beta-like cells only, however in this case it will be necessary to generate antibodies against mutant protein.
Round 2
Reviewer 2 Report
Question 3 has not been addressed: Please explain how is it possible that individual cells express only one type of mRNA, when all contain both WT and mutant alleles.
Author Response
Question 3 has not been addressed: Please explain how is it possible that individual cells express only one type of mRNA, when all contain both WT and mutant alleles.
Human beta-like cells differentiated from iPSCs express both WT and mutant alleles. As we wrote in 2.3 of the Results section " Since differentiated mutant cells produced insulin, we decided to verify the existence of the predicted splice isoform along with wild type mRNA in the patient's beta-like cells. We extracted RNA from isogenic beta-like cell lines and control H9 cells, converted to cDNA, amplified the region around mutation, cloned and sequenced." cDNAs were cloned in E.coli and each E.coli clone that we sequenced contained only one insulin sequence mutant or wild type. ". In cDNA library from edited MNDINSi001-AE61 cells we found only wild type INS transcript indicating that gene editing led to the homozygous restoration of the INS mRNA, wherein in cDNA library from patient’s beta-like cells there were clones containing the insertion of 29 bp between neighboring exons and the ones containing only wild type fragment of the mRNA (Figure 4B)." To calculate an approximate level of the mutant gene transcription we sequenced 32 E coli clones from the cDNA library "To compare the ratio of the wild type mRNA of the INS gene and the mutant one we sequenced 32 clones from the patient's beta-like cells library. We found out that 68% of clones contained wild type cDNAs, while 32% of clones contained mutant sequence with the 29 bp insertion. "
To make it clear for the readers we changed the last sentence "To compare the ratio of the wild type mRNA of the INS gene and the mutant one we sequenced 32 E.coli clones from patient's beta-like cells cDNA library. We found out that 68% of clones contained wild type cDNAs, while 32% of clones contained mutant sequence with the 29 bp insertion.